# Cardiopulmonary bypass in a rat model may shorten the lifespan of stored red blood cells by activating caspase-3

Lu Han[1,2☯], Lianlian Li[3☯], Hangya Linghu[4], Lei Zheng[5], Daming Gou[2]*

1 Department of Anesthesiology, Affiliated Hospital of Zunyi Medical University, Zunyi, Guizhou, China, 2 Department of Anesthesiology, KweiChow Moutai Hospital, Renhuai, Guizhou, China, 3 Department of Anesthesiology, Hospital of Banan District, Chongqing, China, 4 Department of Anesthesiology, Bishan Maternity and Child Hospital of Chongqing, Chongqing, China, 5 Department of Anesthesiology, Affiliated Hospital of Guizhou Medical University, Guiyang, Guizhou, China

☯ These authors contributed equally to this work.
* gdmzy@yeah.net

## Abstract

**Data Availability Statement:** All relevant data are within the manuscript and its Supporting information files.

### Background

Red blood cell transfusion is required for many types of surgery against cardiovascular disease, and the function of transfused cells appears to decline over time. The present study examined whether transfusion also reduces red blood cell lifespan in a rat model.

### Material and methods

Bypass in rats were established by connecting a roll pump to the femoral artery and vein. Then FITC-labeled stored red blood cells from rats were transfused in the animals, and the cells in circulation were counted after transfusion. In separate experiments, stored red blood cells were incubated with bypass plasma *in vitro*, and the effects of incubation were assessed on cell morphology, redox activity, ATP level, caspase-3 activity, and phosphatidylserine exposure on the cell surface. These *in vivo* and *in vitro* experiments were also performed after pretreating the stored red blood cells with the caspase-3 inhibitor Z-DEVD-FMK.

### Results

Bypass significantly decreased the number of circulating FITC-labeled stored red blood cells and increased the proportions of monocytes, neutrophils and splenic macrophages that had phagocytosed the red blood cells. *In vitro*, bypass plasma altered the morphology of red blood cells and increased oxidative stress, caspase-3 activity and phosphatidylserine exposure, while decreasing ATP level. Pretreating stored red blood cells with Z-DEVD-FMK attenuated the effects of bypass on caspase-3 activity, but not oxidative stress, in stored red blood cells.

**Funding:** This work was supported by funds from the National Natural Science Foundation of China (81860083). The funders had no role in study design, data collection and analysis, decision to publish, or preparation of the manuscript.

**Competing interests:** The authors have declared that no competing interests exist.

## Discussion

Bypass appears to shorten the lifespan of stored red blood cells, at least in part by activating caspase-3 in the cells.

## Introduction

On-pump cardiac surgery, one of the main therapies for cardiovascular diseases, requires more blood transfusion than other types of surgeries [1–3]. Within a few days after transfusion, the donated red blood cells are known to lose function, as measured in terms of elongation index [4] and 2,3-diphosphoglycerate concentration [5]. Whether the lifespan of red blood cells shortens after transfusion is unclear, and this is important to clarify in order to optimize transfusion volumes and post-transfusion patient management.

We examined this question in the present study by focusing on two processes known to affect red blood cells after transfusion during bypass surgery: phagocytosis by neutrophils and macrophages, which recognize phosphatidylserine on the surface of red blood cells [6, 7]; and apoptosis, a suicide pathway in which lack of ATP, excessive oxidation and/or the presence of pro-inflammatory factors in the environment activate cysteine-containing aspartate-specific protease (caspase) 3 [8–10]. We explored these processes by creating cardiopulmonary bypass in a rat model that involves systemic inflammation similar to that in bypass patients [11].

## Materials and methods

### Animals and reagents

Male Sprague-Dawley rats (320 ± 20 g) were purchased from Chengdu Dashuo Lab Animals (Chengdu, China). The animals were housed in a temperature-controlled room maintained at 22–24˚C and 45–50% relative humidity, under a 12-h light/dark cycle. They had *ad libitum* access to food and water. The rats were allowed to acclimate to their new surroundings for three days before experiments. Animal experiments were approved by the Ethics Committee of the Affiliated Hospital of Zunyi Medical University, China [KLLY(A)-2019-012].

Commercial kits were purchased to assay the following: reactive oxygen species (ROS; Beyotime Biotechnology, Shanghai, China); malonic dialdehyde (MDA), superoxide dismutase (SOD) and reduced L-glutathione (GSH; Nanjing Jiancheng Bioengineering Institute, Nanjing, China); adenosine triphosphate (ATP; Jianglai Biotechnology, Shanghai, China); caspase-3 activity (Abcam, Cambridge, UK); and phagocytic activity of splenic macrophages (Haoyang Biological Manufacture, Tianjin, China). Tissue homogenate rinse solution, 70-μm cell sieves, 10-mL centrifuge tube, monocyte separation solution and cleaning solution were from Haoyang Biological Manufacture (Tianjin, China). Red blood cells lysate was obtained from Solarbio (Beijing, China).

### Rat model of coronary bypass

We established a rat model of coronary bypass as described [11]. Briefly, animals were anesthetized with 2% pentobarbital sodium (50 mg·kg$^{-1}$) and fixed on the heating plate. Then tracheal intubation was performed, and the tracheal tube was connected to a ventilator (Mini Vent, Harvard Apparatus, Boston, MA, USA) at a respiratory rate of 60 times min$^{-1}$, tidal volume of 15 mL·kg$^{-1}$, inhalation-to-exhalation ratio of 1:2.5 and FiO$_2$ of 99%.

The animals were heparinized intravenously (3 mg·kg$^{-1}$), then the right femoral artery and vein were exposed, needles with respective gauges of 20 or 22 were inserted (Braun, Melsungen, Germany), and the needles were connected to tubes and a roll pump (Jostra HL20,

Maquet Cardiopulmonary, Munich, Germany). The left femoral artery was also exposed in order to allow monitoring of arterial blood pressure using a biological signal acquisition system (model no. 4C501H, MedLab, Nanjing, China).

The roll pump drove blood from the femoral artery to femoral vein at a flow rate of 40 ml·kg$^{-1}$·min$^{-1}$ for 2 h, simulating cardiopulmonary bypass. During this period, anal temperature was maintained at 35–37˚C with a heating blanket, and mean arterial pressure was maintained at 60–90 mmHg. The extracorporeal circulation pipeline was transferred for 2 h in the bypass group, while it remained stationary in the sham group.

## Preparation of stored red blood cells and fluorescent labeling

Blood (10 mL) was harvested from the heart of rats into a centrifuge tube (Becton Dickinson, New York, NY, USA), anticoagulated using heparin (15 IU per mL of blood), and centrifuged at 1000 $g$ for 7 min. The sedimented red blood cells (approximately 6 mL) were gently resuspended with 2 mL of blood preservation solution (89 mM sodium citrate, 17 mM citric acid, 177 mM glucose, 1 mM adenine and 19 mM sodium dihydrogen phosphate), then stored at 4˚C in a bag for 5 days.

The specific method by which the stored red blood cells were labeled with fluorescein isothiocyanate (FITC; Meilunbio, Dalian, China) is general membrane labeling by cell tracker. Cells were centrifuged at 2200 $g$ for 5 min at room temperature, and the pelleted cells were washed twice with phosphate-buffered saline (PBS) and incubated for 20 min at 37˚C with FITC at a final concentration of 70 μg/mL. Cells were centrifuged and resuspended twice in PBS to remove free dye.

## *In vivo* experiments with the rat bypass model

FITC-labeled red blood cells were infused into bypass and sham-bypass rats (10% of blood volume) throughout the 2-h bypass period. In some experiments, the FITC-labeled red blood cells were pretreated for 3 h with either 20 μM Z-DEVD-FMK (MCE, Newark, NJ, USA) in dimethylsulfoxide (DMSO; Solarbio, Beijing, China) or the corresponding volume of DMSO, then transfused into animals during bypass. The inhibitor concentration was selected based on the literature [12]. At 2, 24 and 48 h after transfusion, blood samples were drawn from the tail vein into heparinized syringes in order to determine the percentage of stored red blood cells that had been engulfed by monocytes and neutrophils. Some rats were observed for 24 h, then they were sacrificed using sodium pentobarbital and their spleen was harvested in order to determine the percentage of stored red blood cells that had been cleared by splenic macrophages.

## *In vitro* experiments with the rat bypass model

Stored red blood cells (5–12×10$^9$ cells in 0.95 mL) were treated for 3 h with 50 μL 400 μM Z-DEVD-FMK or the same volume of DMSO, then incubated at 37˚C with 0.5 mL of plasma taken from rats after 2 h of bypass or sham-bypass. The plasma had been isolated by harvesting blood and centrifuging it at 2200 $g$ for 5 min. Blood samples were harvested at 2, 24 and 48 h and analyzed to assess the effects of incubation on cell morphology, redox activity, ATP level, caspase-3 activity, and phosphatidylserine exposure on the cell surface.

## Morphology of stored red blood cells

After 2, 24 and 48 h of incubation with plasma (see "*In vitro* experiments" above), red blood cells were isolated by centrifugation, diluted 1:100 with PBS and analyzed for abnormal

morphology on glass slides under a confocal microscope (N-SIM S, Nikon, Tokyo, Japan). Abnormal cells were identified as echinocytesor spherocytes. The percentage of abnormal cells among 500 cells counted in five random fields was determined.

## Caspase-3 activity in stored red blood cells

After 2, 24 and 48 h of incubation with plasma (see "*In vitro* experiments" above), red blood cells (10 μL) were centrifuged at 3000 $g$ for 5 min. The pelleted cells were resuspended in 50 μL of chilled cell lysis buffer (Solarbio), incubated on ice for 10 min, then centrifuged at 10,000 $g$ for 1 min. The supernatant was mixed with an equal volume of reaction buffer containing 10 mM dithiothreitol and substrate DEVD-pNA (Abcam, Cambridge, UK) at a final concentration of 200 μM, the mixture was incubated at 37°C for 1.5 h, and absorbance was measured at 400 nm. Negative control reactions did not contain substrate.

## Phosphatidylserine exposure on the surface of stored red blood cells

After 2, 24 and 48 h of incubation with plasma (see "*In vitro* experiments" above), red blood cells (10 μL) were centrifuged at 3000 $g$ for 5 min, washed twice with PBS, and resuspended in 1 mL of 1× binding buffer (Becton Dickinson, Franklin, NJ, USA). An aliquot (100 μL, $1×10^6$ cells) was incubated with APC-annexin V (5 μL; Becton Dickinson) at room temperature for 15 min in the dark. Cells were washed twice with PBS, then analyzed on a Calibur flow cytometer (Beckman, Fullerton, CA, USA). At least 50,000 events were analyzed using Flowjo software version 7.6.2 (Tree Star, Ashland, OR, USA).

## Phagocytosis of stored red blood cells

After 2, 24 and 48 h of incubation with plasma (see "*In vivo* experiments" above), blood was harvested, 1 mL was mixed with 3 mL of red blood cell lysate (Solarbio), and the mixture was incubated on ice for 20 min, during which it was gently vortexed twice. The resulting clear liquid was centrifuged at 450 $g$ for 10 min at 4°C. The pelleted leukocytes were resuspended in 2 mL of red blood cell lysate, and the mixture was incubated on ice for 10 min, then centrifuged at 450 $g$ for 10 min at 4°C. The pellet was resuspended in 1 mL PBS, and 100 μL was analyzed on a Calibur flow cytometer. At least 10,000 events were analyzed. The percentages of neutrophils and monocytes that had phagocytosed FITC-labeled stored red blood cells were calculated based on in-house flow cytometry and on the total numbers of neutrophils and monocytes determined using an automated blood analyzer (Dymind, Shenzhen, China).

At 24 h after bypass or sham-bypass (see "*In vivo* experiments" above), rats were sacrificed by intraperitoneal anesthesia overdose, the left lower abdomen was dissected, and the spleen was removed and separated from the outer membrane and fat in PBS at 4°C. The spleen was cut into pieces of 1–2 mm$^3$ using ophthalmic forceps, and the pieces were ground up with a rod and passed through a 70-μm cell sieve using tissue homogenate rinse solution.

The sieved tissue homogenate was centrifuged at room temperature at 450 $g$ for 10 min, and the supernatant was discarded. Separation solution was prepared by adding 1 ml of "monocyte separation solution 1" followed by 3 ml of "separation solution 2" into a specially designed 10-mL centrifuge tube. Spleen tissue suspension (2 mL) was layered onto the surface of the separation solution, then the tube was centrifuged at 750 $g$ at room temperature for 30 min, leading to six layers. The layer containing macrophages (second from top) was carefully removed, washed with 5 mL cleaning solution, and centrifuged at 300 $g$ for 10 min. The pellet was resuspended in PBS and the percentage of macrophages that had phagocytosed FITC-labeled stored red blood cells was determined using flow cytometry based on analysis of 10,000 target cells.

### Statistical analysis

Data were reported as mean ± standard deviation and analyzed using SPSS for Windows 21.0 (IBM, Armonk, NY, USA). Differences between groups were assessed for significance using the independent-samples *t* test or Kruskal-Wallis test. Differences associated with $P < 0.05$ were considered significant.

## Results

### Bypass shortens the lifespan of stored red blood cells

Bypass led to significantly greater phagocytosis of FITC-labeled stored red blood cells by monocytes (Fig 1a) and neutrophils (Fig 1b) at 2 and 24 h afterward, but not 48 h afterward. Bypass led to significantly fewer stored red blood cells in circulation at 2, 24 and 48 h (Fig 1c); in fact, the red blood cells were nearly undetectable by 48 h. In addition, we measured the percentage of FITC-labeled stored red blood cells that were engulfed by splenic macrophages. Bypass significantly increased phagocytosis (Fig 1d). Our experiments did not allow us to determine conclusively whether macrophages and neutrophils were phagocytosing intact FITC-labeled red blood cells and/or membrane fragments or microvesicles of labeled red blood cells. In any event, these results suggest that bypass shortens the lifespan of stored red blood cells.

### Bypass plasma activates caspase-3 in stored red blood cells *in vitro*

To explore how bypass may shorten the lifespan of stored red blood cells, we incubated them with plasma from rats that had undergone bypass or sham bypass. We hypothesized that

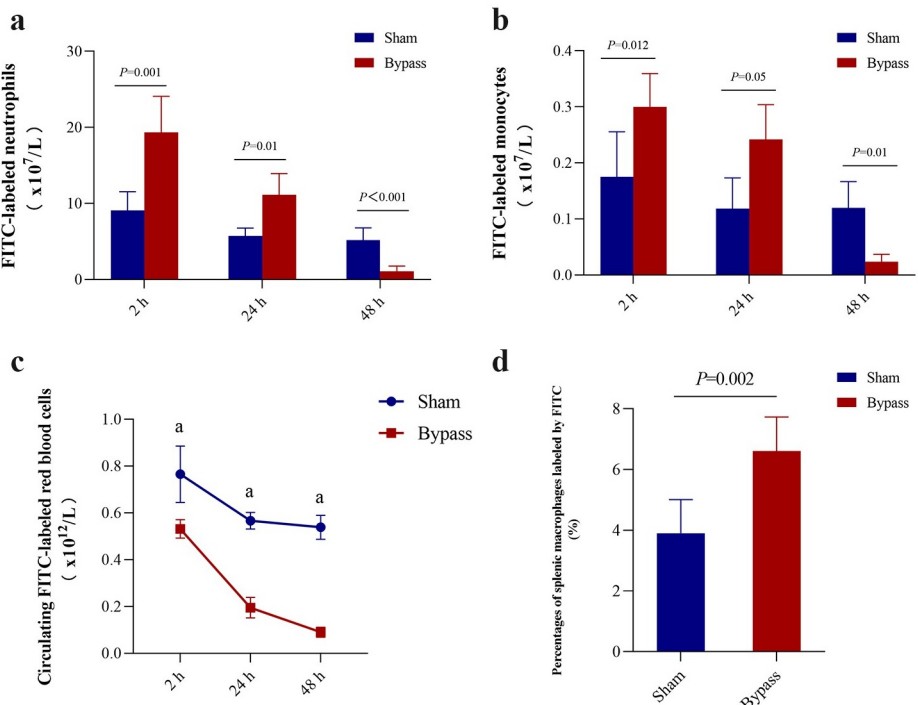

**Fig 1. Bypass shortens the lifespan of stored red blood cells in a rat model of cardiopulmonary bypass. (a, b)** Absolute numbers of (a) monocytes and (b) neutrophils that phagocytosed FITC-labeled stored red blood cells at the indicated time points after 2-h bypass or sham bypass. **(c)** Absolute numbers of FITC-labeled red blood cells over time after bypass or sham bypass. [a] $P < 0.05$ *vs*. bypass group. **(d)** Percentages of splenic macrophages labeled by FITC at 24 h after 2-h bypass or sham bypass. Data are mean ± SD (n = 6 animals per condition).

bypass would induce the production of ROS in stored red blood cells, which would activate caspase-3 and trigger their apoptosis. Indeed, incubating stored red blood cells in plasma from bypass animals significantly increased their levels of ROS and MDA (Fig 2a–2c) and reduced levels of GSH and SOD (Fig 2d and 2e), while activating caspase-3 (Fig 2f).

As additional indicators of cellular damage, we assessed the morphology and ATP levels of the red blood cells. Plasma from bypass animals significantly increased the proportion of stored red blood cells showing abnormal morphology (Fig 3a and 3b), while decreasing intracellular levels of ATP (Fig 3c). Phosphatidylserine exposed on the surface of cells serves as a signal for suicidal red blood cells death or eryptosis in order to clear damaged cells from the circulation [9, 12]. We also found that plasma from bypass animals significantly increased the amount of phosphatidylserine exposed on the surface of stored red blood cells, based on assay of the proportion of red blood cells that bound annexin V (Fig 3d).

To examine whether bypass-induced activation of caspase-3 could explain the observed shorter lifespan of stored red blood cells, we pretreated them with the caspase-3 inhibitor

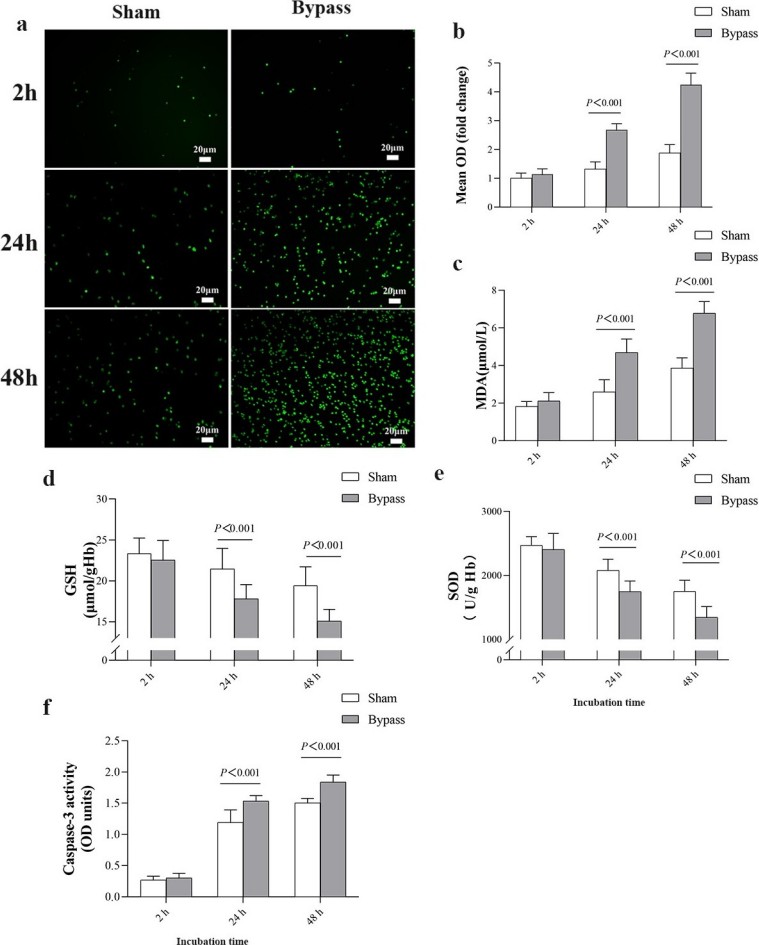

**Fig 2. Bypass induces oxidative damage and activates caspase-3 in stored red blood cells.** Cells were incubated *in vitro* for the indicated times with plasma from rats subjected to bypass or sham bypass. **(a)** Representative micrographs of the suspensions of red blood cells in plasma. Reactive oxygen species (ROS) were stained using DCFH-DA (green). **(b-e)** Levels or activity of (b) ROS, (c) malonic dialdehyde (MDA), (d) l-glutathione (GSH) and (e) superoxide dismutase (SOD) were assayed in the suspensions. (f) Activity of caspase-3 in the suspensions. Data are mean ± SD (n = 10 animals per condition). OD, optical density.

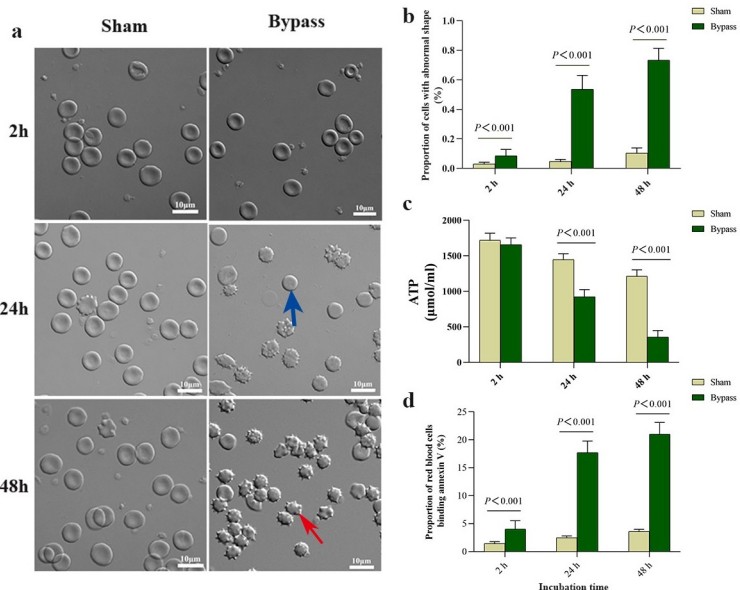

**Fig 3. Bypass alters the morphology of red blood cells, reduces their ATP levels and increases the exposure of phosphatidylserine (PS) on their surface.** Cells were incubated *in vitro* for the indicated times with plasma from rats subjected to bypass or sham bypass. **(a, b)** Representative micrographs and quantification of abnormal morphology among red blood cells in the suspensions. The blue arrow indicates a spherocyte; the red arrow, an echinocyte. Magnification, 1000×. **(c)** Levels of ATP within the red blood cells. **(d)** The proportion of Annexin V-binding red blood cells. Data are mean ± SD (n = 10 animals per condition).

Z-DEVD-FMK before mixing them with plasma from rats in bypass. Cells were destroyed by high concentration of the vehicle (DMSO). To examined whether the concentration of DMSO in our experiments affects the outcomes or the survival of red cells, we incubated rat blood cells (after 5-day storage) in culture medium containing 0.9% normal saline or DMSO for 2 h, then mixed them for 48 h with bypass plasma. We found that levels of caspase-3, ROS, MDA, GSH and SOD were similar between both types of cultures during 48 h (S1a–S1e Fig). The same was observed for cellular morphology, ATP levels and extent of PS exposure (S1f–S1h Fig). We conclude that DMSO exerted negligible effects on red blood cells in our *in vitro* experiments. The inhibitor efficiently inhibited caspase-3 activity in red blood cells (Fig 4a), while minimally affecting their oxidative stress (Fig 4b–4f). It reversed the effects of bypass plasma on abnormal morphology (Fig 5a and 5b), on ATP production (Fig 5c), and on the proportion of red blood cells that bound annexin V (Fig 5d). These results suggest that bypass may shorten the lifespan of stored red blood cells by activating caspase-3-mediated apoptosis.

## Inhibiting caspase-3 prolongs the lifespan of stored red blood cells after bypass

Next we asked whether inhibiting caspase-3 activity would prolong the lifespan of stored red blood cells. FITC-labeled cells were pretreated with Z-DEVD-FMK or DMSO, then transfused into animals subjected to bypass. Although numbers of FITC-labeled monocytes or neutrophils were similar between groups (Fig 6a and 6b), Z-DEVD-FMK pretreatment led to significantly higher numbers of stored red blood cells in circulation at 24 and 48 h (Fig 6c), and to lower percentages of splenic macrophages labeled by FITC (Fig 6d). These results suggest that inhibiting caspase-3 can prolong the lifespan of stored red blood cells *in vivo*.

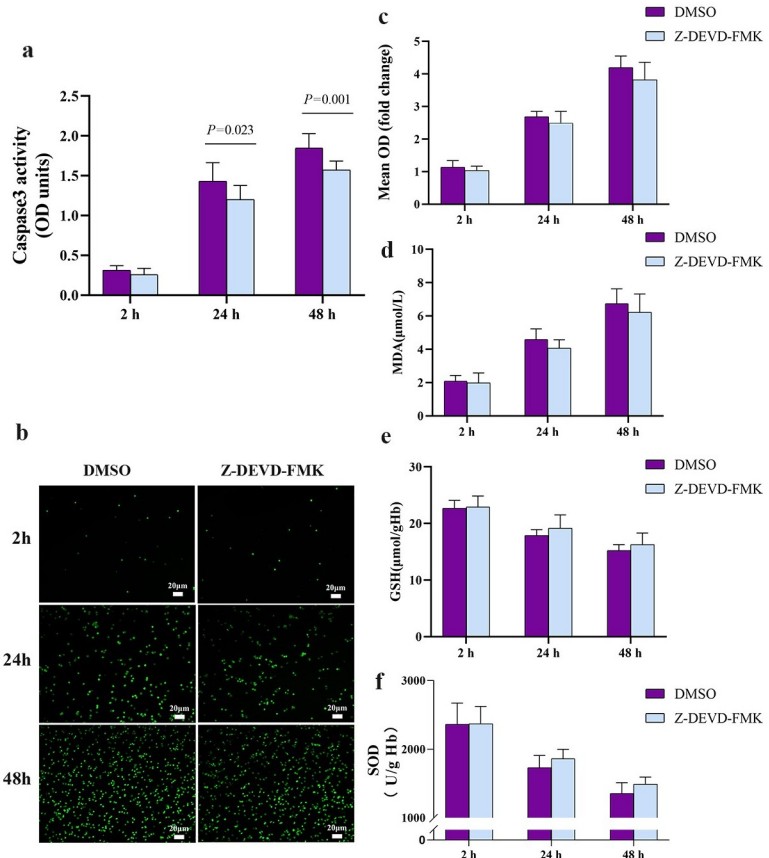

**Fig 4. Inhibiting caspase-3 did not improve oxidative stress in red blood cells.** Cells were pretreated with caspase-3 inhibitor Z-DEVD-FMK or vehicle (DMSO), then mixed with plasma from rats that underwent bypass. **(a)** The inhibitor Z-DEVD-FMK inhibited caspase-3 in stored red blood cells. **(b)** Representative micrographs of the suspensions of red blood cells in plasma. Reactive oxygen species (ROS) were stained using DCFH-DA (green). **(c-f)** Levels or activity of (b) ROS, (c) malonic dialdehyde (MDA), (d) L-glutathione (GSH) and (e) superoxide dismutase (SOD) were assayed in the suspensions. Data are mean ± SD (n = 10 animals per condition).

## Discussion

Using a rat model of bypass to simulate the procedure applied in many types of cardiac surgery, we found that more than half of stored red blood cells were cleared from the circulation within only 24 h after transfusion. Our experiments *in vivo* and *in vitro* suggest that bypass activates caspase-3-mediated apoptosis in the transfused red blood cells, which is associated with greater phagocytosis by monocytes, neutrophils and splenic macrophages. Our results further suggest that inhibiting caspase-3 can at least partly reverse these negative effects of bypass.

The red blood cells in our experiments were stored for 5 days before transfusion, which is comparable to 10 days of storage for human red blood cells [13, 14], and 14 days is the average duration of storage before use in the clinic [15, 16]. We found that the tranfused cells were rapidly cleared from the circulation, such that few were detectable by two days after bypass. Our results are consistent with previous studies showing that up to 30% of transfused red blood cells are rapidly hemolyzed or removed from the circulation [17, 18]. Consistent with our assays involving stored red blood cells, experiments with freshly collected cells also showed

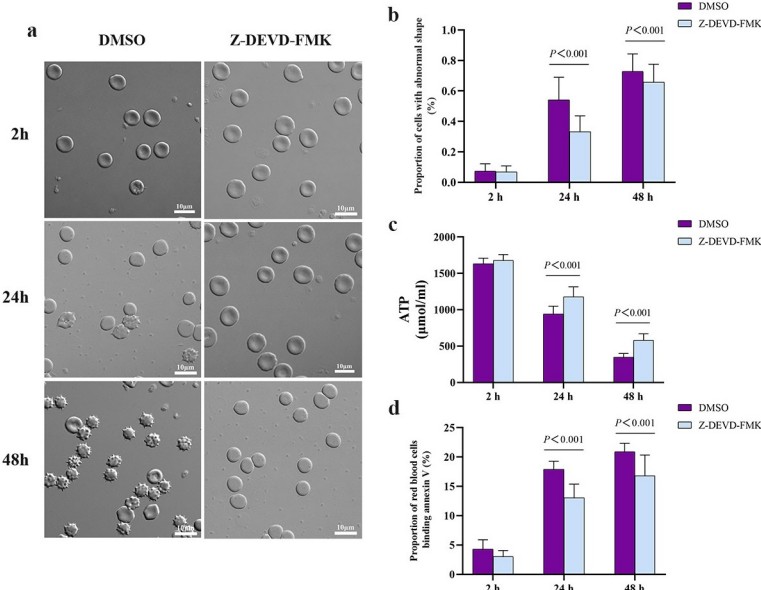

**Fig 5. Inhibiting caspase-3 in stored red blood cells blocks damage induced by bypass.** Cells were pretreated with caspase-3 inhibitor Z-DEVD-FMK or vehicle (DMSO), then mixed for the indicated times with plasma from rats that underwent bypass. **(a, b)** Representative micrographs and quantification of abnormal morphology among red blood cells in the suspensions. Magnification, 1000×. **(c)** Levels of ATP within the red blood cells. **(d)** The proportion of red blood cells that bound annexin V. Data are mean ± SD (n = 10 animals per condition).

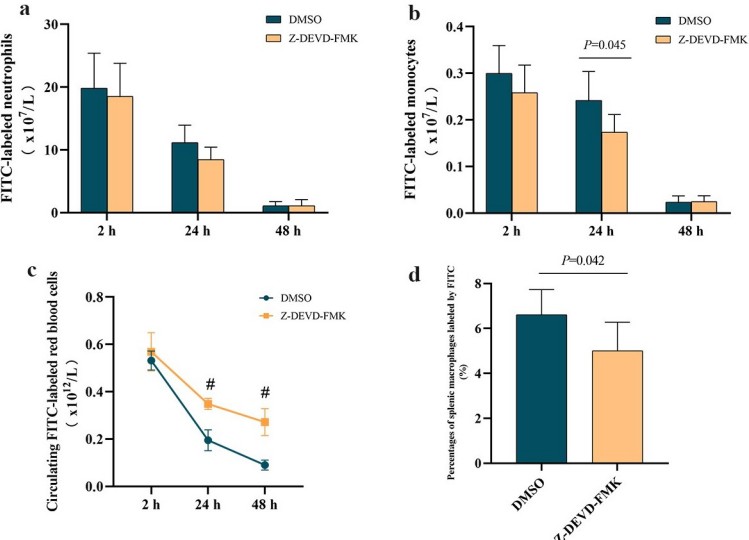

**Fig 6. Inhibiting caspase-3 in stored red blood cells before bypass prolongs their lifespan after transfusion.** FITC-labeled cells were pretreated with caspase-3 inhibitor Z-DEVD-FMK or vehicle (DMSO), then transfused into rats that underwent bypass. **(a, b)** Numbers of (a) monocytes and (b) neutrophils that phagocytosed FITC-labeled stored red blood cells at the indicated time points after 2-h bypass. **(c)** Absolute numbers of FITC-labeled stored red blood cells in circulation at different times after bypass. [#]$P<0.001$ *vs*. Z-DEVD-FMK group. **(d)** Percentages of splenic macrophages labeled by FITC at 24 h after bypass. Data are mean ± SD (n = 6 animals per condition).

that oxidative stress activated caspase-3, leading to lower ATP levels and higher the amount of phosphatidylserine exposed on the surface of stored red blood cells [12].

Bypass may shorten the lifespan of stored red blood cells by stimulating the production of inflammatory cells and cytokines as a result of ischemia-reperfusion and endotoxemia [10]. This inflammatory response may expose red blood cells to oxidative injury, which in turn activates caspase-3 [12, 19]. Consistent with these ideas, we found that bypass increased levels of ROS and MDA as well as activity of caspase-3 in stored red blood cells. However, inhibiting caspase-3 did not alleviate oxidative stress.

ATP is fundamental for maintaining red blood cell morphology and integrity [20]. Our results showed that bypass decreased the ATP content in red blood cells and increased the incidence of dysmorphism. In addition, if ATP levels in red blood cells are inadequate, $Ca^{2+}$ channels cannot prevent intracellular accumulation of the ion, which leads to exposure of phosphatidylserine and phosphatidylethanolamine on the surface, which macrophages recognize as "eat-me" signals [20]. Here we found that bypass increased the proportion of annexin V-binding red blood cells *in vitro* and the percentage of stored red blood cells engulfed by macrophages *in vivo*. These results indicate that bypass induces damage of red blood cells.

One of the key steps in damaging red blood cells is the activation of caspase-3. In red blood cells, caspase-3 regulates externalization of phosphatidylserine and phagocytosis of oxidatively stressed red blood cells, and it cleaves band 3, which helps link the membrane to the spectrin-based skeletal network [12, 21, 22]. We examined whether we could reverse the damaging effects of bypass on transfused red blood cells using Z-DEVD-FMK, which specifically inhibits caspase-3 and has been widely used tissue injury models *in vitro* and *in vivo* [21, 23]. We found that bypass activated caspase-3 in red blood cells, which Z-DEVD-FMK partially reversed *in vitro*. The inhibitor Z-DEVD-FMK only partially reduced the destruction of red blood cells *in vivo*: the number of red blood cells in circulation still decreased within 48 h after transfusion. This suggests that there may be additional pathways, independent of caspase-3, that contribute to destruction of transfused red blood cells. For example, free hemoglobin in the bloodstream as well as the activated complement system can attack red blood cells [24, 25]. In addition to activating caspase3, oxidative stress or deficiency of the antioxidant system may increase the uptake of $Ca^{2+}$ by cells through activation of cation channels, causing red blood cells damage [26]. Oxidative stress can also cause red blood cells to be cleared by the body by altering the conformation of the band-3 protein on the red blood cells membrane and by prompting some receptors on the erythrocyte surface, including CD47 and platelet-reactive protein-1 receptors, to bind to ligands [27, 28].

This study demonstrated that Z-DEVD-FMK, by inhibiting caspase-3, may protect red blood cells from the damaging effects of bypass. However, our findings should be interpreted with caution given that our *in vitro* model may not accurately capture all aspects of transfusion in the clinic. Nevertheless, the model allows detailed molecular and cellular analyses of what happens to red blood cells after transfusion, and we were able to validate our major *in vitro* findings with experiments *in vivo*. Future experiments could verify our findings by examining whether anti-oxidants reverse the damaging effects of bypass, and whether bypass damages fresh red blood cells to the same extent as it damages stored cells; if so, we could exclude free hemoglobin in the bloodstream as a contributor to the observed damage. We conducted our experiments with bulk red blood cells, yet red blood cells can differ substantially in lifespan, and bypass may affect the subpopulations to different extents [29]. Future work should consider fractionating red blood cells by lifespan and examining the effects of bypass on each subpopulation individually.

Our experiments suggest potential protective effects of Z-DEVD-FMK, which has already been shown to enhance stored red blood cells function and survival. Whether this compound

can be used in the clinic is unclear, since little appears to be known about its potential toxicity in humans [30]. Certainly its potential for protecting the function of transfused blood products deserves further study.

## Conclusion

Our results suggest that activation of caspase-3 is a major driver of damage to stored red blood cells after transfusion during on-pump cardiac surgery. Inhibiting this enzyme in transfused red blood cells and/or blocking the downstream apoptosis and phagocytosis of these cells may reduce the transfusion volumes needed during on-pump cardiac surgery and thereby improve prognosis.

## Supporting information

**S1 Fig. Stored red blood cells were incubated with DMSO or normal saline (NS), then incubated in bypass plasma.** At the indicated time points, the suspension was assayed for (a) caspase-3, (b) reactive oxygen species (ROS), (c) malonic dialdehyde (MDA), (d) L-glutathione (GSH), (e) superoxide dismutase (SOD), (f) abnormal morphology, (g) ATP, and (h) ability to bind annexin V. Data are mean ± SD (n = 10 animals per condition). OD, optical density. (TIF)

**S1 Table. The minimal underlying data set about the manuscript.** (ZIP)

## Acknowledgments

The authors gratefully acknowledge the assistance of Professor Jie Zhang (West China Hospital, Sichuan University and Key Laboratory of Transplant Engineering and Immunology, Ministry of Health of China) with confocal microscopy.

## Author Contributions

**Data curation:** Lu Han.

**Methodology:** Lu Han, Lianlian Li, Hangya Linghu, Lei Zheng.

**Resources:** Lu Han.

**Software:** Lu Han.

**Writing – original draft:** Lu Han.

**Writing – review & editing:** Lu Han, Daming Gou.

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
