## [Decision Letter · Decision Letter 0]

5 Apr 2023

PONE-D-23-00813

Cardiopulmonary bypass in a rat model may shorten the lifespan of stored red blood cells by activating caspase-3

PLOS ONE

Dear Dr. Han,

Thank you for submitting your manuscript to PLOS ONE. After careful consideration, we feel that it has merit but does not fully meet PLOS ONE’s publication criteria as it currently stands. Therefore, we invite you to submit a revised version of the manuscript that addresses the points raised during the review process.

Please address address the reviewer comments, which are offered to improve the quality of the paper. Suggestions regarding additional data or experiments are not required for publication.

We look forward to receiving your revised manuscript.

Kind regards,

Heather Faith Pidcoke, MD, MSCI, PhD

Academic Editor

PLOS ONE

Journal Requirements:

"This work was supported by funds from the National Natural Science Foundation of China (81860083). The authors are grateful to the participants for their interest and cooperation. "

"This work was supported by funds from the National Natural Science Foundation of China (81860083). The funders had no role in study design, data collection and analysis, decision to publish, or preparation of the manuscript."

3. "Please amend the manuscript submission data (via Edit Submission) to include author "Lianlian Li , Hangya Linghu, Lei Zheng , Daming Gou"

Reviewers' comments:

Reviewer's Responses to Questions

**Comments to the Author**

1. Is the manuscript technically sound, and do the data support the conclusions?

Reviewer #1: Partly

Reviewer #2: Partly

2. Has the statistical analysis been performed appropriately and rigorously? 

Reviewer #1: Yes

Reviewer #2: Yes

3. Have the authors made all data underlying the findings in their manuscript fully available?

Reviewer #1: Yes

Reviewer #2: Yes

4. Is the manuscript presented in an intelligible fashion and written in standard English?

Reviewer #1: No

Reviewer #2: Yes

5. Review Comments to the Author

Reviewer #1: In this manuscript a rat blood bypass model was used to observe effects of 2 hours of pumping blood through the bypass roller pump on stored and transfused red cells (RBCs). Stored red cells were labelled with FITC, infused into the animal and pumped for 2 hours. The presence of labeled RBCs in circulation was measured at 2, 24 and 48 hrs post infusion. Presence of fluoresce associated with other cells was measured at the same time points. To determine the mechanism of reduced RBC life span, plasma from animals that had been on bypass was incubated in vitro with stored, FITC -labelled RBCs and changes in the RBCs were measured at 2, 24, and 48 hours. RBCs stored in bypass plasma exhibited changes associated with apoptosis such as caspace activation, phosphatidylserine expression and ATP decline. Pre-treatment of RBCs with a caspace inhibitor prior to infusion into rats with by-pass reduced the loss of red cells. Plasma from such experiments, when incubated with RBCs did not induce the apoptosis-related changes which occurred with plasma from animals that received RBCs not treated with the caspace inhibitor. The authors concluded that reduced lifespan of infused stored red cells in animals on by-pass was due to free radical generation during the by-pass phase which activated RBCs caspace and induced these cells to go through apoptosis.

Comments

This is an interesting report on how RBCs could be protected by pretreatment with a caspase inhibitor from damage due to by-pass pumps. There are several questions that come to mind:

These experiments utilize stored red cells which are likely to have developed storage lesions and be more fragile than fresh cells. Would freshly collected red cells produce the same results in these experiments? What storage time of human red cells does 10 days of rat red cell storage represent? This could be mentioned in the discussion.

Free radicals have a very brief life span. How is it possible that plasma from by-pass animals could retain the free radicals? If the free radicals remain in the plasma it should be possible to include anti-oxidant molecule(s) that would block the free radical effect on red cells when they are incubated with the by-pass plasma. Such an experiment would strengthen the free radical argument. Alternatively, there could be other explanations such as complement activation and/or free hemoglobin release. These too could be measured in the by-pass plasma. Similarly performing the experiments with fresh cells which may be less prone to lysis would reduce the possibility that free hemoglobin is involved in the responses.

Phagocytosis of red cells

FITC labels only the outside of red cells. Physical stress/damage from the by-pass pumps likely leads to release red cell membrane fragments or microvesicles while the red cell remains in circulation. In several places it is stated that macrophages and neutrophils can internalize damaged RBCs because there is increased fluorescence associated with these cells. An alternate explanation could be that these cells are internalizing membrane fragments or microvesicles labelled with FITCs or that the fragments and microvesicles are attached to the outside of the cells. Fluorescent microscopy should be able to determine which process is going on.

The manuscript has many grammatical errors. A review by a qualified English writer would be recommended.

Reviewer #2: In this research article Han et al. examined the effects of cardiopulmonary bypass on the recovery and post-transfusion physiology of RBCs. By using a rat model of cardiopulmonary bypass to simulate the clinical procedure applied in cardiac surgery patients they found that bypass increases the erythrophagocytosis and thus, reduces the recovery of stored RBCs. Moreover, by using in vitro experiments which included incubation of stored cells with plasma isolated by bypass or sham bypass animal blood, they detected oxidative stress, caspase-3 activation, cell morphology distortions, externalization of phosphatidylserine and reduced intracellular ATP levels in reconstitutions with bypass plasma. Finally, inhibition of caspase-3 activity seems to prevent in part those negative effects, leading to prolong RBC lifespan after transfusions to rats subjected to bypass.

It is an interesting work and a well-written manuscript. I congratulate the authors especially for the Materials and Methods section which is described in detail. I highlight here, however, some concerns and points that need improvement:

Major comments

- The statement that “the function of transfused cells appears to decline over time” (L19) or “after surgery” (L46) and other similar statements present throughout the manuscript are not literally true. We know that the recovery of stored RBCs varies as a function of numerous donor- and recipient-related factors and that a percentage of transfused cells are removed from the circulation the first 24h post transfusion. However, the “survivals” circulate for a normal period in the recipient. Moreover, 24h recovery is actually the most measured property of transfused RBCs per se. In contrast, other physiological measures of transfused RBCs (deformability, morphology, fragility etc) are only indirectly assessed in vivo by measurements performed in the mixed RBC population of the recipient which consists of donor and recipient cells. We must take into consideration that these different cell populations may mutually change as a result of the transfusion event, that also includes residual donor plasma and transmittance of biological response modifiers, such as extracellular vesicles. In vitro models, like the one used by the authors, can reveal transfusion effects exclusively on donor RBCs but only those related to body temperature and recipient plasma. Despite useful and informative, these models are not equal to a “real” transfusion event.

-Figure 4 results (L242): In the basic experiment reported in the text (L239-243) stored RBC pretreated with caspase-3 inhibitor and then incubated with bypass or sham bypass plasma, while Fig 4 shows another comparison, namely that between inhibitor vs. DMSO treated RBCs with bypass plasma. First of all, DMSO seems to significantly affect the caspase activity (Fig. 2f vs. Fig. 4a). Moreover, ROS staining shown in the representative micrographs of Fig. 4b is inconsistent with the ROS bar graphs shown in Fig. 4c. And even for the comparison DMSO vs. inhibitor, one would expect lower ROS and MDA levels in the inhibitor-treated cells (vs. the DMSO-treated ones) in the context of significantly lower caspase activity (Fig. 4a). However, this is not the case as shown in Fig. 4c-f. Having these in mind, the authors cannot support that the inhibitor reversed the effects of bypass plasma on the ROS, MDA, GSH and SOD levels as reported in L242-243. On the opposite, the results shown n Fig. 5 are consistent with lower shape abnormalities and PS exposure on cells that exhibit lower caspase activation.

-according to the results L271-273 the inhibition of caspase-3 cannot reverse all the negative effects of bypass as reported in the discussion L296-297. Please rephrase.

-L341-342: Beyond doubt, there are additional pathways involved in RBC destruction (including the band-3/IgG/C3b and CD47-related surface removal signaling) and the authors should report them along with the respective citations.

Minor comments

-Introduction is too short.

- A study limitations paragraph is missing.

- In my opinion, red arrow in Fig. 3 indicates an echinocyte and not an acanthocyte.

-There is a mismatch in the a-f labeling between the Figure 4 plates/graphs and the Legend.

- The two sentences in L271-274 need rephrasing.

- “It has been shown that caspase-3 is present in red blood cells…” Another work to support this statement in stored RBCs is that of Kriebardis et al. (TRANSFUSION 2007;47:1212-1220).

6. PLOS authors have the option to publish the peer review history of their article (what does this mean?). If published, this will include your full peer review and any attached files.

Reviewer #1: No

Reviewer #2: No

---

## [Author Response · Author response to Decision Letter 0]

15 Jun 2023

11 June 2023

Dear Dr Pidcoke:

We would like to thank you for the time spent on our manuscript “Cardiopulmonary bypass in a rat model may shorten the lifespan of stored red blood cells by activating caspase-3” (PONE-D-23-00813). Below we provide point-by-point responses to referees’ comments, and we have marked the major revisions in the manuscript in blue.

We hope the manuscript can now be judged suitable for publication in the PLOS ONE. We would be happy to respond to any further questions or comments that you or the reviewers may have.

Sincerely,

Daming Gou

Department of Anesthesiology

KweiChow Moutai Hospital 

Renhuai, Guizhou, China

Tel: +86-13985219675

E-mail: gdmzy@yeah.net

RESPONSES TO COMMENTS FROM EDITOR:

Response: We have revised our manuscript according to the PLOS ONE's style requirements.

"This work was supported by funds from the National Natural Science Foundation of China (81860083). The authors are grateful to the participants for their interest and cooperation. " 

"This work was supported by funds from the National Natural Science Foundation of China (81860083). The funders had no role in study design, data collection and analysis, decision to publish, or preparation of the manuscript."

Response: We have updated our Funding Statement.

3. "Please amend the manuscript submission data (via Edit Submission) to include author "Lianlian Li , Hangya Linghu, Lei Zheng , Daming Gou"

Response: We have revised the manuscript submission data. 

4. We note that you have included the phrase “data not shown” in your manuscript. Unfortunately, this does not meet our data sharing requirements. PLOS does not permit references to inaccessible data. We require that authors provide all relevant data within the paper, Supporting Information files, or in an acceptable, public repository. Please add a citation to support this phrase or upload the data that corresponds with these findings to a stable repository (such as Figshare or Dryad) and provide and URLs, DOIs, or accession numbers that may be used to access these data. Or, if the data are not a core part of the research being presented in your study, we ask that you remove the phrase that refers to these data

Response: We have deleted the phrase “data not shown” in our manuscript.

Response: We have added the full ethics statement in the ‘Methods’ section of our manuscript file (lines 69-71).

Response: We have added supporting information S1 table as minimal data set to support our results described in manuscript.

Response: We have reviewed and corrected the relevant literatures.

RESPONSES TO COMMENTS FROM REVIEWER 1: 

1. These experiments utilize stored red cells which are likely to have developed storage lesions and be more fragile than fresh cells. Would freshly collected red cells produce the same results in these experiments? What storage time of human red cells does 10 days of rat red cell storage represent? This could be mentioned in the discussion.

Response: We now mention that the 5-day storage period of the rat red blood cells in our experiments is comparable to 10-day storage of human red blood cells, which is comparable to the average duration of storage before use in the clinic (line 301-303). 

2. Free radicals have a very brief life span. How is it possible that plasma from by-pass animals could retain the free radicals? If the free radicals remain in the plasma it should be possible to include anti-oxidant molecule(s) that would block the free radical effect on red cells when they are incubated with the by-pass plasma. Such an experiment would strengthen the free radical argument. Alternatively, there could be other explanations such as complement activation and/or free hemoglobin release. These too could be measured in the by-pass plasma. Similarly performing the experiments with fresh cells which may be less prone to lysis would reduce the possibility that free hemoglobin is involved in the responses.

Response: We thank the reviewer for these insightful comments, which we suggest as important hypotheses to examine in future work (lines 335-337 and lines 348-351). 

3. Phagocytosis of red cells FITC labels only the outside of red cells. Physical stress/damage from the by-pass pumps likely leads to release red cell membrane fragments or microvesicles while the red cell remains in circulation. In several places it is stated that macrophages and neutrophils can internalize damaged RBCs because there is increased fluorescence associated with these cells. An alternate explanation could be that these cells are internalizing membrane fragments or microvesicles labelled with FITCs or that the fragments and microvesicles are attached to the outside of the cells. Fluorescent microscopy should be able to determine which process is going on.

Response: In our hands, FITC labels both the membrane and cytoplasm of red cells (Figure 1, below). We agree with the reviewer that the macrophages and neutrophils may have acquired FITC labeling by phagocytosing entire red blood cells or membrane fragments or microvesicles of red blood cells. We now mention this possibility on line 200-203. 

Figure 1. Representative fluorescence micrographs from our experiments, in which red blood cells in suspension were labeled with FITC. The area boxed in red on the left is shown at higher magnification at right.

RESPONSES TO COMMENTS FROM REVIEWER 2:

1. The statement that “the function of transfused cells appears to decline over time” (L19) or “after surgery” (L46) and other similar statements present throughout the manuscript are not literally true. We know that the recovery of stored RBCs varies as a function of numerous donor- and recipient-related factors and that a percentage of transfused cells are removed from the circulation the first 24h post transfusion. However, the “survivals” circulate for a normal period in the recipient. Moreover, 24h recovery is actually the most measured property of transfused RBCs per se. In contrast, other physiological measures of transfused RBCs (deformability, morphology, fragility etc) are only indirectly assessed in vivo by measurements performed in the mixed RBC population of the recipient which consists of donor and recipient cells. We must take into consideration that these different cell populations may mutually change as a result of the transfusion event, that also includes residual donor plasma and transmittance of biological response modifiers, such as extracellular vesicles. In vitro models, like the one used by the authors, can reveal transfusion effects exclusively on donor RBCs but only those related to body temperature and recipient plasma. Despite useful and informative, these models are not equal to a “real” transfusion event.

Response: We appreciate the limitations that the reviewer points out in our modeling approach, which we have highlighted in the limitations section of the Discussion (lines 352-353). Nevertheless, we believe that our transfusion model is valuable for elucidating molecular and cellular details of the events affecting the lifespan of transfused red blood cells, and we were careful to validate our findings through in vivo experiments. 

2. Figure 4 results (L242): In the basic experiment reported in the text (L239-243) stored RBC pretreated with caspase-3 inhibitor and then incubated with bypass or sham bypass plasma, while Fig 4 shows another comparison, namely that between inhibitor vs. DMSO treated RBCs with bypass plasma. First of all, DMSO seems to significantly affect the caspase activity (Fig. 2f vs. Fig. 4a). 

Moreover, ROS staining shown in the representative micrographs of Fig. 4b is inconsistent with the ROS bar graphs shown in Fig. 4c. And even for the comparison DMSO vs. inhibitor, one would expect lower ROS and MDA levels in the inhibitor-treated cells (vs. the DMSO-treated ones) in the context of significantly lower caspase activity (Fig. 4a). However, this is not the case as shown in Fig. 4c-f. Having these in mind, the authors cannot support that the inhibitor reversed the effects of bypass plasma on the ROS, MDA, GSH and SOD levels as reported in L242-243. 

On the opposite, the results shown Fig. 5 are consistent with lower shape abnormalities and PS exposure on cells that exhibit lower caspase activation.

Response: We thank the reviewer for careful consideration of our data, which led us to detect a mistake in the original Figure 4. We have now provided the correct image in the revised manuscript and revised line 244-246 to read “To examine whether bypass-induced activation of caspase-3 could explain the observed shorter lifespan of stored red blood cells, we pretreated them with the caspase-3 inhibitor Z-DEVD-FMK before mixing them with plasma from rats in bypass”. 

In response to the reviewer’s comment, we incubated rat blood cells (after 5-day storage) in culture medium containing 0.9% normal saline or DMSO for 2 h, then mixed them for 48 h with bypass plasma. We found that levels of caspase-3, ROS, MDA, GSH and SOD were similar between both types of cultures during 48 h (Figure 2a-e, below). The same was observed for cellular morphology, ATP levels and extent of PS exposure (Figure 2f-h, below). We conclude that DMSO exerted negligible effects on red blood cells in our in vitro experiments. 

We agree with the reviewer that Z-DEVD-FMK minimally affected oxidative stress in red blood cells, as shown in the revised Figure 4b-f. We have revised the text accordingly (lines 254-255).

Figure 2. Stored red blood cells were incubated with DMSO or normal saline (NS), then incubated in bypass plasma. At the indicated time points, the suspension was assayed for (a) caspase-3, (b) reactive oxygen species (ROS), (c) malonic dialdehyde (MDA), (d) L-glutathione (GSH), (e) superoxide dismutase (SOD), (f) abnormal morphology, (g) ATP, and (h) ability to bind annexin V. Data are mean ± SD (n = 10 animals per condition). OD, optical density.

3. According to the results L271-273 the inhibition of caspase-3 cannot reverse all the negative effects of bypass as reported in the discussion L296-297. Please rephrase.

Response: We have revised the sentence to “Our results further suggest that inhibiting caspase-3 can, at least partly, reverse these negative effects of bypass” (line 299-300).

4. Title: L341-342: Beyond doubt, there are additional pathways involved in RBC destruction (including the band-3/IgG/C3b and CD47-related surface removal signaling) and the authors should report them along with the respective citations.

Response: We now mention these pathways and corresponding references (line 337-342). 

Minor comments

1. Introduction is too short.

Response: We have extended the introduction with relevant background.

2. A study limitations paragraph is missing.

Response: We have added discussion of study limitations (lines 343-355).

3. In my opinion, red arrow in Fig. 3 indicates an echinocyte and not an acanthocyte.

Response: We thank the reviewer for this careful insight. We have changed the annotation to “echinocyte”.

4. There is a mismatch in the a-f labeling between the Figure 4 plates/graphs and the Legend.

Response: As noted above, we have revised the Figure 4 and corresponding legend.

5. The two sentences in L271-274 need rephrasing.

Response: We have revised these sentences (lines 276-281).

6. “It has been shown that caspase-3 is present in red blood cells…” Another work to support this statement in stored RBCs is that of Kriebardis et al. (TRANSFUSION 2007; 47:1212-1220).

Response: We have added this as ref 23 (lines 454-457).

---

## [Decision Letter · Decision Letter 1]

7 Aug 2023

Cardiopulmonary bypass in a rat model may shorten the lifespan of stored red blood cells by activating caspase-3

PONE-D-23-00813R1

Dear Dr. Gou,

We’re pleased to inform you that your manuscript has been judged scientifically suitable for publication and will be formally accepted for publication once it meets all outstanding technical requirements. Please address the minor revisions suggested by Reviewer 2 below before submitting the final version of your manuscript.

Kind regards,

Heather Faith Pidcoke, MD, MSCI, PhD

Academic Editor

PLOS ONE

Additional Editor Comments (optional):

Reviewers' comments:

Reviewer's Responses to Questions

**Comments to the Author**

1. If the authors have adequately addressed your comments raised in a previous round of review and you feel that this manuscript is now acceptable for publication, you may indicate that here to bypass the “Comments to the Author” section, enter your conflict of interest statement in the “Confidential to Editor” section, and submit your "Accept" recommendation.

Reviewer #1: All comments have been addressed

Reviewer #2: All comments have been addressed

2. Is the manuscript technically sound, and do the data support the conclusions?

Reviewer #1: Yes

Reviewer #2: Yes

3. Has the statistical analysis been performed appropriately and rigorously? 

Reviewer #1: Yes

Reviewer #2: Yes

4. Have the authors made all data underlying the findings in their manuscript fully available?

Reviewer #1: Yes

Reviewer #2: Yes

5. Is the manuscript presented in an intelligible fashion and written in standard English?

Reviewer #1: Yes

Reviewer #2: Yes

6. Review Comments to the Author

Reviewer #1: (No Response)

Reviewer #2: Please report the specific method/kit by which the stored RBCs were labeled with FITC (surface Ab conjugated to FITC? General membrane labeling by cell tracker or another reagent?) (L. 107)

Please rephrase the paragraph of Morphology of stored RBCs (“Abnormal cells were identified as acanthocytes or spherocytes”. L138) according to the text of Fig. 3 legend (“The blue arrow indicates a spherocyte; the red arrow, an echinocyte”. L241)

7. PLOS authors have the option to publish the peer review history of their article (what does this mean?). If published, this will include your full peer review and any attached files.

Reviewer #1: No

Reviewer #2: No

---

## [Editor Report · Acceptance letter]

12 Sep 2023

PONE-D-23-00813R1 

Cardiopulmonary bypass in a rat model may shorten the lifespan of stored red blood cells by activating caspase-3 

Dear Dr. Gou:

I'm pleased to inform you that your manuscript has been deemed suitable for publication in PLOS ONE. Congratulations! Your manuscript is now with our production department. 

Kind regards, 

on behalf of

Dr. Heather Faith Pidcoke 

Academic Editor

PLOS ONE